# Battery Lifespan of an Implantable Middle Ear Device

Luca Bruschini [1,2], Francesca Forli [1,2,*], Giacomo Fiacchini [1,2], Rachele Canelli [1,2], Stefano Berrettini [1,2,3] and Francesco Lazzerini [1,4]

1. Otolaryngology, Audiology and Phoniatrics Unit, University of Pisa, 56126 Pisa, Italy
2. Department of Surgical, Medical and Molecular Pathology and Critical Care Medicine, University of Pisa, 56126 Pisa, Italy
3. Clinical Science, Intervention and Technology, Karolinska Institute, 17177 Stockholm, Sweden
4. Department of Clinical and Experimental Medicine, University of Pisa, 56126 Pisa, Italy
* Correspondence: francesca.forli@gmail.com

**Abstract:** Background: The Carina system (Cochlear Ltd., Sydney, Australia) is a totally implantable device providing acoustic amplification in adult patients with moderate-to-severe sensorineural or mixed hearing loss. One of the main concerns about such a totally implantable device has been represented by the subcutaneous battery lifespan. The aim of this article is to report the analysis of battery performances in a series of Carina-implanted patients after a long follow up. Methods: In this retrospective study, the technical data of a series of patients implanted with the Carina middle ear implant in our clinic have been analysed, extracting the data from the log of telemetric measures. Results: The mean lifespan cutback was 0.43 h/years (from 0 to 0.71 h/year), with a strong negative significant correlation between the follow-up period and the percentage of battery residual lifespan. Conclusion: The lifespan of the Carina's battery seems consistent with the manufacturer statement of a pluri-decennial lifespan, avoiding the need of an early surgical substitution and providing a full day of use of the system even after up to 12 years from the implantation.

**Keywords:** active middle ear implant; battery; totally implantable hearing aid

## 1. Introduction

Totally implantable hearing devices are indicated as a treatment for moderate-to-severe hearing loss in patients who can not benefit from conventional hearing aids [1].

In particular, the Carina system (Cochlear Ltd., Sydney, Australia) is a totally implantable device providing acoustic amplification in adult patients with sensorineural hearing loss (SNHL) or mixed hearing loss (MHL) [2]. All the components of the device, the battery, processor, transducer, and microphone, are implanted under the skin. So, this device substantially is invisible to others. The same transducer is used for the totally implantable system and for a semi-implantable system, the Middle Ear Transducer (MET) [3–5]. It received a "Conformité Européene" (CE) mark in 2006 for SNHL and in 2007 for MHL. Further, even if the reliability of such implants has been an object of concern, it has been demonstrated how an evolution in the technology has provided a good improvement in consistency over the newer generation of devices [2].

The Carina is nowadays indicated for adult patients with moderate-to-severe SNHL or MHL with an audiometric threshold of 30–85 dB, especially in the high frequencies [2,6,7]. Candidates should try conventional hearing aids with unsuccessful results. Patients with recurrent outer ear canal infections, who cannot wear conventional hearing aids, are also candidates for the Carina system [8].

The Carina device was used in patients with ossicular chain malformations and conductive hearing loss. In these cases, the device must impart its mechanical energy to the cochlea via pathways other than the normal middle ear conductive pathway, such as the round window or the stapes footplate after a stapedotomy. In this regard, Siegert et al.,

in 2007, implanted a device in subjects with congenital auricular atresia, using a modified transducer system, with satisfactory results, whereas Tringali et al., in 2008, successfully treated a case of severe conductive hearing loss by directly stimulating the stapes footplate with a MET™ V transducer for conductive applications, in a child with Franceschetti syndrome and bilateral auricular atresia associated with middle ear malformation [9]. Moreover, Didczuneit-Sandhop and Langer consider appropriate the carina implant for the treatment of otosclerosis [10].

Exclusion criteria are a history of recurrent middle ear infections or a known middle ear malformation, inner ear disorders, retro-cochlear or central hearing impairment [8].

Even if other middle ear implants can be placed through minimally invasive approaches, such as the exclusive transcanal approach, the usual surgical technique for Carina implantation involves a posterior atticotomy after a 3–5 cm post-auricular incision [11,12].

Even if there is no consensus on the optimal placement of the subcutaneous microphone, three possible locations sites are convenient for the positioning: anteriorly and superiorly to the external auditory canal, posteriorly to the external auditory canal, and on the mastoid tip [12].

Even if this system has been recently discontinued in some countries, it has been previously confirmed as a valid alternative to hearing aids, especially in patients that are unable to wear the traditional hearing prosthesis for psychological problems or external auditory canal issues.

All the main components of the Cochlear Carina are implanted under the skin, including the microphone, the sound processor, and even the rechargeable batteries. The subcutaneous microphone collects sound from behind the ear and sends it to the processor. The signal, then, is amplified, processed, and forwarded to the actuator as mechanical movement. The transducer can be coupled to the stapes, oval window, or round window.

Through a telemetric arrangement, the programming system permits the fitting of the implanted device, similarly to a cochlear implant. The charging system has a base station, a charging coil, and a battery charger. To charge the battery of the system, the charging coil is placed over the skin in magnetic contact with the implanted device. A full charging takes approximately 45 min. Through a remote control, the recipient can turn the system on and off and adjust the volume [5].

One of the main concerns about such a totally implantable device has been represented by the subcutaneous battery lifespan.

The battery, indeed, is charged with a coil placed on the skin over the implant. The procedure should be performed daily and it lasts about 1 to 1.5 h. The manufacturer asserts that, as new, the battery could last more than 30 h with a single cycle of charge [13,14].

With normal use, then, the battery span should decrease gradually. From the manufacturer testing, indeed, considering a daily use of at least 16 h, the Carina's battery lifespan is supposed to be 25–30 years, with a gradual decrement of performance [14] (see Figure 1).

Then, since the battery is implanted under the skin, a substitution would require a surgical intervention for the electronic capsule replacement. The middle ear transducer, however, should not be removed during the battery change procedure.

The aim of this article is to report the analysis of battery performances in a series of Carina-implanted patients, submitted to surgery, and followed up in the Otolaryngology Audiology and Phoniatric Operative Unit of Pisa University Hospital, defining the useful lifespan of the subcutaneous batteries and assessing a mean rate of the performance's decrement. The results, then, will be compared with those reported from the manufacturer from experimental accelerated testing.

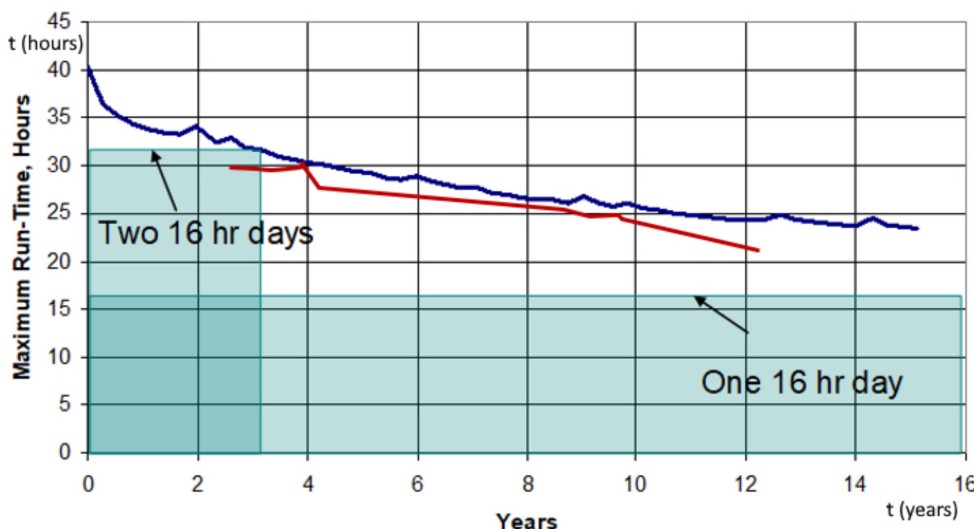

**Figure 1.** A scheme showing the Carina's projected battery lifespan through the years. Figure reported and modified from the Memo Battery Testing Summary, LIB-00385—Cochlear whitepaper. The blue line is the house-made Carina projection, while the red line is our data.

## 2. Materials and Methods

Technical data of a series of patients implanted with the Carina middle ear implant in a national referral centre for otosurgery from 2009 to 2019 were retrospectively reviewed.

Then, the mean dB gain with Carina, comparing the pure tone average (PTA) (250–2000 Hz) of a free field audiometry with and without the system switched on, was calculated.

The subjective benefit of the intervention was assessed collecting the Glasgow Benefit Inventory (GBI) questionnaire [15] for each patient.

All the recipients received the third generation of the Carina prosthesis (so called G3+T1) or newer versions. All the implants, therefore, own the same type of battery.

The follow up was calculated as the period between the activation and the last telemetric measure in months.

The data were extracted from the log of telemetric measures. All the measures were ruled out by the same specialist with the same software, released by the manufacturer.

The percentage of residual battery lifespan was calculated as 100% of battery health corresponding to a single cycle charge duration of 30 h. The single charge lasting decrement was expressed as a percentage.

The data obtained were analysed using the Pearson correlation through SPSS statistics for Windows, Version 23.0 (IBM Corp. in Armonk, NY, USA). $p$ values < 0.05 were considered as statistically significant.

## 3. Results

Since 2009, 19 patients were submitted to Carina implantation. One of those patients received a bilateral implantation; therefore, the number of Carina systems implanted in our centre was 20.

In all cases, the device was implanted through a surgical procedure under general anaesthesia. The surgery was conducted through a slightly curved post-auricular incision, two flaps, cutaneous and muscular, to visualise the spine of Henle and the mastoid region. So, a small superior atticotomy was created, about 2 cm wide, to expose the body of the incus. The arm of the mounting bracket of the device was modified to perfectly place the device on the incus. The mounting bracket was then fitted securely to the mastoid cortex using bone screws. Bone beds for the device were drilled posterior and superior to the atticotomy. The transducer was mounted in the retaining ring. The tip of the transducer was advanced into the hole on the incus and the positioning was evaluated using the TLA. The device was placed on the bone bed and the microphone was placed under the skin in the retroauricular region.

Eight of the implanted patients attend regular annual follow-up procedures, by mean telemetric measures, fitting and audiological evaluations. The lost-to-follow-up patients was due to:

(a) Explantation for extrusion after prosthesis infection in two cases;
(b) Explantation for patient's wish in one case;
(c) A stopping of prosthesis use, due to hearing loss progression over time and subsequent lack of benefit from the amplification through the implanted device in three cases;
(d) Moving to other centres for follow up in two cases (one case with bilateral implant);
(e) Death of the recipient in one case;
(f) Unknown reasons in two cases.

The mean follow-up time of the study population was 91.25 months (7.6 years, from 31 to 148 months).

The mean postoperative functional gain with the device switched on was 27.5 dB (12.50–36.25 dB).

The mean GBI total result was 30.9 (21.0–38.7). For the General subsection of the GBI, the mean result was 35.5 (25.5–42.8); for the Social Support subsection, the mean result was 5.8 (0.5–11.3); for the Physical subsection, the mean result was 5.2 (0.3–10.0).

The mean lifespan percentage of the battery was 86.88% (from 71% to 100%), equivalent to an estimated mean maximum single cycle of charge duration of 26.06 h (from 21.3 to 30 h).

The mean lifespan cutback was 0.43 h/years (from 0 to 0.71 h/year).

It was possible to find a strong negative significant correlation between the follow-up period and the percentage of residual battery lifespan (Pearson Coefficient = $-0.970$, $p < 0.001$) (see Figure 2).

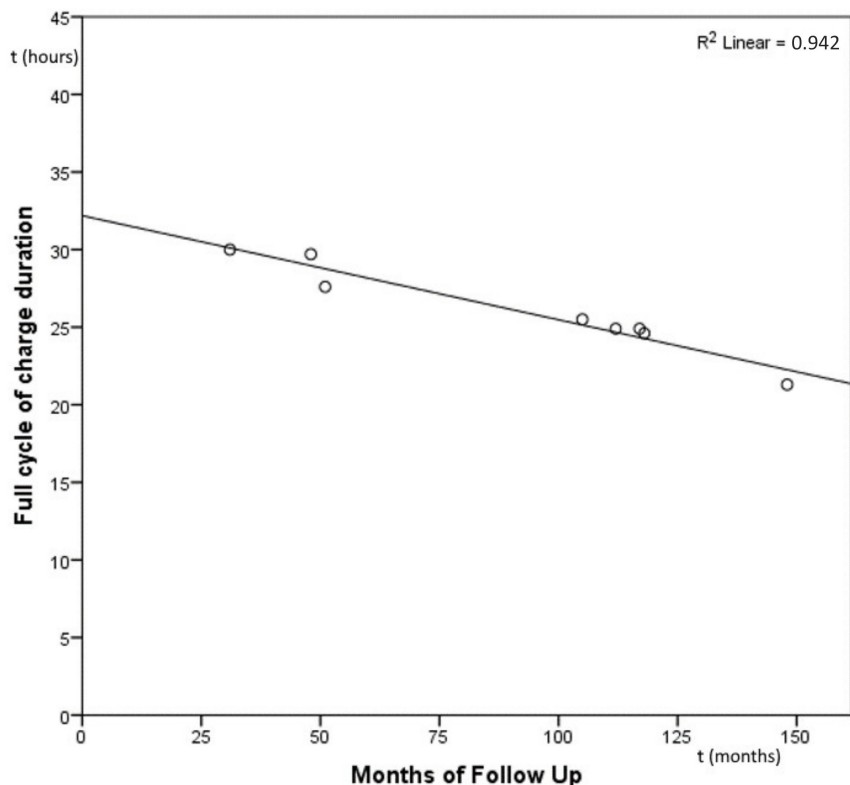

**Figure 2.** Scatterplot plotting showing a a strong negative significant correlation between the follow-up period and the percentage of residual battery lifespan (Pearson Coefficient = $-0.970$, $p < 0.001$).

## 4. Discussion

The advantages of a totally implantable hearing aid are several: e.g., a higher acceptance by the user, as it would be an invisible prosthesis; the prevention of atmosphere

exposition, leading to a lower rate of damage from temperature and moisture; the protection from water, permitting a full aquatic experience; and the comfortable use while sleeping and moving or doing exercise, preventing damage from dislodgement and perspiration [16,17].

The Carina system is a fully implantable middle ear prosthesis. It was previously reported that this system can provide a substantial benefit in improving hearing performances in different environmental conditions [2,12], valid functional gain, and significant improvement of speech perception. Moreover, the recipients reported a high degree of satisfaction, with additional advantages related to the cosmetic aspects, and the possibility of greater freedom in performing common daily activities [12,18].

The published audiological results showed a good hearing gain using the Carina system, with a mean functional gain of 24–29 dB [12,19]. In our study population of Carina system recipients, implanted at our clinic, the hearing results are in line with those reported in literature, with a mean gain of 27.5 dB with the implant switched on.

In addition, the subjective benefit in terms of quality-of-life perception, assessed with a GBI questionnaire, confirm what was reported in literature, with a mean total post-implant result of 30.9. This value, indeed, is in line with what was previously reported by many authors, confirming a good subjective satisfaction from middle ear implantation surgery [20].

However, a fully implantable device presents some drawbacks. Firstly, the subcutaneous microphone can present an attenuation of signals and reduced sensitivity because of the above skin flap, and it can record issue-transmitted sounds such as blood flow, muscular contractions, or the user's own voice [17,21]. A careful surgical procedure can limit those drawbacks [22], and most of the Carina recipients in this study rarely reported persistent feedback from the implantable microphones.

In all our cases, the surgical procedure was performed through general anaesthesia, in about 3 h, without any relevant surgical complication in any of the patients. In the patients reported in the present paper, we did not record any significant postoperative variation in hearing thresholds, for either air or bone conduction thresholds, with a mean deterioration of the thresholds of circa 3 dB, indicating the absence of surgical damage to the cochlea. A very useful tool for the surgeon, certainly, is the transducer loading assistant (TLA), i.e., the telemetric system, created by the parent company, to evaluate the correct position and the coupling of the transducer. The TLA makes possible the excellent results of the device. Moreover, in order to improve the evaluation of the coupling of the device, Cebulla et al. recommends the use of the intraoperative auditory evoked potentials in place of the TLA system [23].

Further, a totally implantable system requires an implantable battery capable of a full day of energy supply that can be recharged rapidly wirelessly and with a sufficient lifespan that permits the avoidance of further surgeries to back it up; moreover, it should not heat significantly, and it must be safe [16,24–26].

The Carina's battery is a 50 mAh lithium-ion battery. Like all batteries with lithium-ion technology, it goes through a certain decrement of performance related to the use and the subsequent cycles of charge. The manufacturer declares a useful battery lifespan of 25 to 30 years, assuming a daily use of the implant of at least 16 h. After that period or whenever the mean duration of a full cycle of charge does not permit a full-day utilization, the Carina's implanted battery should be substituted surgically. These data were calculated from the manufacturer, according to lab testing procedures of accelerated life-testing, and supposing a complete discharge of the battery at every cycle [14]. According to these data, the expected annual lifespan decrement was around 1 h, over a period of 14 years (see Figure 1).

Over their lifespan, lithium-ion batteries tend to degrade gradually [27]. A variety of chemical and mechanical mechanisms are related to the normal degradation of lithium-ion batteries, but the temperature is strongly implicated in the reduction in performance over time [28]. It can be speculated that the temperature under the skin, where the Carina battery is placed in vivo, may differ from the temperature of the experimental setting. Furthermore,

batteries can generate heat when being charged or discharged, especially at high currents, and the dissipation of the heat in a subcutaneous space can be problematic. Moreover, charging Li-ion batteries beyond 80% can drastically accelerate battery degradation [27] and, even if the manufacturer testing protocol contemplated a full charge and discharge of the battery at every cycle, batteries are not always fully charged or discharged in real-life in vivo use. This could bring discrepancies between the testing protocol and in vivo results.

Thanks to the telemetric system, it was possible to estimate the battery lifespan of our cohort of Carina recipients. In our study population, after a mean follow-up period of 91.25 months (7.6 years), the mean lifespan percentage of the batteries was 86,88%, corresponding to a mean estimated duration of a single cycle of charge of 26 h, with an annual lifespan cutback of 0.43 h (see Figure 1). Those results show how, even in the patient with the longer follow up (up to 12 years), it is still possible to use the Carina system all day long.

Further, our data seem to confirm the experimental finding of the manufacturer, with a progressive decrement of the batteries' performances, but with the capacity to guarantee enough run-time hours for an all-day long use of the system for decades.

Furthermore, despite the need for some explantations and some patients lost to follow up in our casuistry, none of these adverse events were related to battery issues.

In our opinion, the present article can add knowledge in the field of totally implantable hearing devices, since it is the only paper in scientific literature that reports pluri-decennial in vivo results of totally implantable batteries.

The limitations of our study are related to the relative paucity of the casuistry, even due to the considerable amount of lost-to- follow-up patients.

## 5. Conclusions

The Carina middle ear implant system, despite being recently discontinued in some countries, represents a valid amplification system for adult patients with moderate-to-severe sensorineural or mixed hearing loss. Despite some drawbacks that are common in totally implantable devices, we can state from the present study that the batteries are safe, and their lifespan is good, providing a full day of use of the system even after up to 12 years from the implantation.

In our study population, with a significant follow-up period, none of the patients needed a revision surgery for a too-low battery status or battery failures.

The lifespan of the Carina's battery, even in vivo, seems to stick to the manufacturer statement of a pluri-decennial lifespan, avoiding the need of an early surgical substitution.

**Author Contributions:** Conceptualisation, L.B. and F.L.; methodology, F.L.; validation, L.B. and S.B.; data curation, F.L., G.F. and R.C.; original draft preparation, F.L. and L.B.; writing—review and editing, F.F.; supervision, L.B. All authors have read and agreed to the published version of the manuscript.

**Funding:** This research received no external funding.

**Institutional Review Board Statement:** The study was conducted according to the guidelines of the Declaration of Helsinki, and approved by the Local Review Board. On behalf of all the authors, I am stating that the Institutional Review Board approved the publication of the paper and, since it is a retrospective study, it does not require Ethical Committee approval.

**Informed Consent Statement:** Informed consent was obtained from the subjects involved in the study.

**Data Availability Statement:** All the reported data are available upon reasonable request.

**Conflicts of Interest:** The authors declare no conflict of interest.

## Abbreviations

This table lists abbreviations and acronyms used in the paper.

CE       Conformité Européene
GBI     Glasgow Benefit Inventory
MET    Middle ear transducer
MHL    Mixed hearing loss
SNHL   Sensorineural hearing loss
TLA     Transducer loading assistant

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
