# Peer review of "Battery Lifespan of an Implantable Middle Ear Device"

_audiolres, doi:10.3390/audiolres12050049_

Round 1

Reviewer 1 Report

The paper is interesting,quite well prepared  but my main doubt is novelty and first of all scientific level. The manuscript is like a case study. I can not find any  scientific novelty. Maybe it is enough to publish a report but for scientific journal deeper study is required.

Detailed remarks:

- affilliation no  4 is not used for any author

- some phrases in Introduction are repeated several times

- some acronyms are not defined

Reviewer 2 Report

Minors. 1. To be improved: Authors write the text in first person. Being a scientific paper, I recommend them to adhere to the concept of impersonally and change the formulation at the third person. 2. Review of the affiliation, I did not understand which of authors has the forth affiliation.  3. Line 122 - reformulation of the phrase (change „thanks” with „through”. 4. Line 179 the phrase and the idea is not complete. 5. Authors should mention about the limitation and the strong point of the article. 5. I suggest authors to introduce a table with abbreviations before references. 

Good points: 1. The method description and results are clearly presented. The introduction part is well written. The conclusion support the results and objectives. 

Reviewer 3 Report

Thanks for the opportunity of reviewing this manuscript.

I only have few minor points:

- were the telemetric measures performed by the same operator and by the same computer and software in all the studied cases?

- in the discussion section, line 179 some text is missing. Please check.
